# Transcriptome analysis of gibberellins and abscisic acid during the flooding response in *Fokienia hodginsii*

**Shunde Su[1,2], Tengfei Zhu**[3], **Jun Su**[3], **Jian Li**[4], **Qing Zhao**[4], **Xiangyang Kang**[1]*,
**Renhua Zheng**[2]*

**1** Beijing Advanced Innovation Center for Tree Breeding by Molecular Design, National Engineering Laboratory for Tree Breeding, Key Laboratory of Genetics and Breeding in Forest Trees and Ornamental Plants, Ministry of Education, College of Biological Sciences and Technology, Beijing Forestry University, Beijing, China, **2** Key Laboratory of Timber Forest Breeding and Cultivation for Mountainous Areas in Southern China Key Laboratory of Forest Culture and Forest Product Processing Utilization of Fujian Province, Fuzhou, China, **3** Basic Forestry and Proteomics Research Center, College of Forestry, Fujian Provincial Key Laboratory of Haixia Applied Plant Systems Biology, Fujian Agriculture and Forestry University, Fuzhou, China, **4** College of Forestry, Fujian Agriculture and Forestry University, Fuzhou, China

\* kangxy@bjfu.edu.cn (XK); zrh08@126.com (RZ)

**Data Availability Statement:** The sequence data reported in this study have been deposited in the Genome Sequence Archive in BIG Data Center, Beijing Institute of Genomics (BIG), Chinese Academy of Sciences, under accession numbers

## Abstract

Flooding is one of the main abiotic stresses suffered by plants. Plants respond to flooding stress through regulating their morphological structure, endogenous hormone biosynthesis, and genetic signaling transduction. We previously found that *Fokienia hodginsii* varieties originating from Gutian exhibited typical flooding tolerance traits compared to three other provenances (Yongzhou, Sanming, Nanping), expressed as increased height, longer diameter at breast height (DBH), and smaller branch angle. Herein, the changes in endogenous gibberellins (GA) and abscisic acid (ABA) contents were measured under flooding stress in *F. hodginsii*, and ABA was found to decrease, whereas GA increased with time. Furthermore, the GA and ABA contents of the varieties originating from Gutian and the three other provenances were measured, and the results indicated that *F. hodginsii* from Gutian could respond more rapidly to flooding stress. The transcriptomes of the varieties originating from Gutian and the other three provenances were compared using RNA sequencing to explore the underlying genetic mechanisms of the flood-resistant phenotypes in *F. hodginsii*. The results indicated that two flood-stress response genes (TRINITY_DN142_c0_g2 and TRINITY_DN7657_c0_g1) were highly related to both the ABA and GA response in *F. hodginsii*.

## Introduction

Water is essential for plant growth, but excess water, such as that caused by flooding, may severely limit plant growth. Under flooding conditions, plants are typically submerged or waterlogged [1, 2], and the resulting $O_2$ deficiency can cause damage [3]. Plants have a variety of adaptive mechanisms to avoid $O_2$ deficiency, such as aerenchyma formation, adventitious root formation, and the control of shoot, petiole, or internode elongation [1]. The adaptation

CRA005262, that are publicly accessible at https://bigd.big.ac.cn/gsa/browse/CRA005262.

**Funding:** the "Eagle Program" of Fujian Province, funded by the Department of Human Resources and Social Security of Fujian Province; 2. the "Fujian Cypress Generation 1 Core Breeding Population Construction Research" (No. 2021R1010004), funded by the Department of Science and Technology of Fujian Province. Role played by the funder in the research: is the one who sets the direction of the research and provides financial support. No author receives a salary from the research project, all funds are used for research.

**Competing interests:** The authors have declared that no competing interests exist.

of plants to long-term flooding also leads to variations in metabolism, hormone content, and enzyme systems [4]. These are considered as powerful drivers of adaptive evolution, leading to a wide range of physiological, molecular, and morphological adaptations [5, 6].

Plants balance the synthesis and transport of plant hormones and regulate the response to waterlogging via complex signaling processes [7–9]. Over the past few decades, numerous studies have shown that ABA and GA have antagonistic effects on many developmental processes in plants, including seed maturation, seed dormancy and germination, root initiation, hypocotyl and stem elongation, and flower transition. In addition, ABA, a well-established stress hormone, plays a key role in plant responses to abiotic stresses such as drought, flooding, salt and low temperature. Interestingly, recent evidence suggests that GA is also involved in plant responses to adverse environmental conditions. Thus, the complex crosstalk network between ABA and GA, mediated by different key regulatory factors, has been extensively investigated and recorded [10]. Under flooding stress, gibberellins (GAs), and abscisic acid (ABA) play the most crucial roles.

The expression of GA is induced directly or indirectly by ethylene. GAs are directly involved in escaping flooding stress in both DELLA (N-terminal D-E-L-L-A amino acid sequence) pathways [11, 12]. Ethylene and ABA are directly responsible for stomatal closure under waterlogging stress, whereas GAs are responsible for stomatal opening. In summary, GA and ABA play influential roles in resisting submergence stress, whereby ABA biosynthesis is reduced and GA signaling is induced for shoot elongation [13–15].

*Fokienia hodginsii* is mainly distributed in the humid climate zone of the central subtropics and is an important constituent species of subtropical evergreen broad-leaved forests and mixed natural forests of coniferous and broad-leaved species [16]. It is cultivated widely within its range as an excellent timber tree species [17]. It takes about 5–7 years from germination to flowering, with flowering in March-April and fruiting in October-November.

We established multi-generation seed orchards by means of genetic techniques, including seed source tests and determination of offspring of superior trees. During this process, we discovered that *F. hodginsii* from Gutian had a distinctive phenotype that differed from the other provenances.

According to historical information, the large-scale construction of reservoirs since 1956 submersed most of the region of Gutian, and the provenance of *F. hodginsii* from Gutian was collected in the late 1970s [18]. Therefore, we hypothesized that the phenotypes of *F. hodginsii* in the Gutian area may have been shaped by flooding stress. This study will reveal, for the first time, the molecular mechanism of flooding resistance-related phenotypes in *F. hodginsii* and enrich the understanding of the molecular basis of flooding resistance in woody plants.

## Materials and methods

### Different provenances of *F. hodginsii*

In this research, the wild type was collected from Forest Park in Fuzhou, Fujian, China (26°4′27.12″N 119°17′49.20″E); the provenance 1–Yongzhou was collected in Hunan, China (26°25′26.25″N 111°36′25.17″E); the provenance 2–Sanming was collected in Fujian, China (117°38′20.40″N 26°15′50.04″E); the provenance 3–Nanping was collected in Fujian, China (118°4′51.29″ N 27°22′57.88″E); and the provenance 4–Gutian was collected in Fujian, China (118°48′35.72″N 26°33′20.19″E).

### Plant growth conditions

The seedling of *F. hodginsii* were grown under hydroponics in nutrient solution (10 g/L Huaduoduo 2# Peters®10-30-20+TE ICL Specialty Fertilizers) for two weeks. The growth

conditions included 22˚C, 16-h light/8-h dark cycle, 60 μmol·m$^2$·s$^{-1}$ photon flux density, and 85% relative humidity.

## Flooding assay

In the flooding stress experiment, the 2-week-old seedlings were completely immersed in purified water for 6 h/12 h/24 h/72 h, and there were three replicates per treatment and 15 samples per replicate, all treatments were performed under complete dark conditions that included 22˚C and 85% relative humidity.

## Simple Sequence Repeat (SSR) assay

Genomic DNA was extracted from the fresh leaves (100mg) of the four provenances of *F. hodginsii* using the cetyltrimethylammonium bromide (CTAB) method [19]. We used the assembled contig data (at least 200 bp) to search for hexanucleotide repeats of SSR loci. Identified SSR loci were then grouped into different classes. The SSR locus density was determined based on the frequency of SSR loci and the total length of contigs containing SSRs. Eleven pairs of SSR primers were used to analyse [20] (S1 Table). We also evaluated the motif length, loci numbers, and mean repeat numbers for the selected repetitive motifs [21].

The polymerase chain reaction (PCR) was conducted in a final volume of 20 μL containing 1 μL genomic DNA, 1 μL each of forward and reserve primer (10 μM), 10 μL of Dream Taq$^{TM}$ Green PCR Master Mix (Thermo Fisher Scientific, USA), and 7 μL ddH$_2$O. The following PCR conditions were used: an initial denaturation of 95˚C for 2 min; 35 cycles of 95˚C for 30 s, annealing temperature of 53˚C for 60 s, and 72˚C for 35 s; followed by a 10-min extension at 72˚C. The amplified products were evaluated on 5% agarose gel using the BIO-RAD Gel Doc$^{TM}$ XR+. Fragment sizes were determined using Image Lab software version 6.0.

## Quantification of endogenous ABA and GA$_3$

The contents of endogenous ABA and GA$_3$ were determined via high-performance liquid chromatography-mass spectrometry (HPLC-MS) [22].

**Metabolites extraction.** The samples were precisely weighed to Eppendorf tubes. After adding two small steel balls and 1000 μL of extraction solution (precooled at -20˚C, acetonitrile-methanol-water, 2:2:1), the samples were vortexed for 30 s, and then were homogenized for 4 min at 40 Hz and sonicated for 5 min in ice-water bath. The homogenate and sonicate circle were repeated three times, followed by incubation at -40˚C for 1 h and centrifugation at 12000 rpm (RCF = 13800(×g), R = 8.6cm) and 4˚C for 15 min. An 80 μL aliquot of the clear supernatant was transferred to an auto-sampler vial for UHPLC-MS/MS analysis.

**Standard solution preparation.** Stock solutions were individually prepared by dissolving or diluting each standard substance to give a final concentration of 10 mmol/L. An aliquot of each of the stock solutions was transferred to a 10 mL flask to form a working standard solution. A series of calibration standard solutions were then prepared by stepwise dilution of this standard solution.

**UHPLC-MRM-MS analysis.** The UHPLC separation was carried out using an ACQUITY UPLC-I/CLASS(Waters), equipped with a Waters ACQUITY UPLC$^®$BEH C18(100 × 2.1 mm, 1.7 μm, Waters). The mobile phase A was 0.1% formic acid in water, and the mobile phase B was methanol. The column temperature was set at 35˚C. The auto-sampler temperature was set at 10˚C and the injection volume was 2 μL. Waters Xevo TQ-S triple quadrupole mass spectrometer (Waters) was applied for assay development. Typical ion source parameters were: Capillary voltages = 3.5kV, Cone voltages = 42V, Desolvation Temperature = 650˚C, Desolvation gas flow = 1000(L/Hr), Cone gas flow = 150(L/Hr), Nebuliser gas flow = 7.0(Bar)

[23]. The MRM parameters for each of the targeted analytes were optimized using flow injection analysis, by injecting the standard solutions of the individual analytes, into the ESI source of the mass spectrometer. Several most sensitive transitions were used in the MRM scan mode to optimize the collision energy for each Q1/Q3 pair. Among the optimized MRM transitions per analyte, the Q1/Q3 pairs that showed the highest sensitivity and selectivity were selected as 'quantifier' for quantitative monitoring. The additional transitions acted as 'qualifier' for the purpose of verifying the identity of the target analytes. Skyline Software were employed for MRM data acquisition and processing.

**Calibration curves.** Calibration solutions were subjected to UPLC-MRM-MS/MS analysis using the methods described above. The y is the peak areas for analyte, and x is the concentration (nmol/L) for analyte. Least squares method was used for the regression fitting. 1/x weighting was applied in the curve fitting since it provided highest accuracy and correlation coefficient (R2). The level was excluded from the calibration if the accuracy of calibration was not within 80% -120%.

**Limit of Detection (LOD) and Limit of Quantitation (LOQ).** The calibration standard solution was diluted stepwise, with a dilution factor of 2. These standard solutions were subjected to UHPLC-MRM-MS analysis. The signal-to-noise ratios (S/N) were used to determine the lower limits of detection (LLODs) and lower limits of quantitation (LLOQs). The LLODs and LLOQs were defined as the analyte concentrations that led to peaks with signal-to-noise ratios (S/N) of 3 and 10, respectively, according to the US FDA guideline for bioanalytical method validation.

## Total RNA extraction, cDNA reverse transcription, and quantitative real-time (qRT) PCR

A polysaccharide polyphenol plant total RNA extraction kit (TIANGEN) and 500 mg leaves were used to extract the total RNA of the samples, extraction procedure according to instructions and a 1.2% agarose gel was used to detect the quality of the RNA and measured the RNA concentration, absorbance value, 260/230, 260/280 before reverse transcription. The strip was clear and non-dispersive, and the brightness of the 28S band was about twice as bright as the 18S band, confirming that the extract could be used for subsequent analysis.

For the RT-qPCR(QuantStudio™ 7 Flex), 1 μg of total RNA was treated with DNase I and used for cDNA synthesis with a YEASEM Hifair®II 1st Strand cDNA Synthesis SuperMix for qPCR Kit (CAT:1123ES60). ACT7 was selected as internal marker gene for real-time fluorescence quantification, and the primers were designed using the Primer 3.0 website. The primers used in this study are listed in S1 Table. The PCR was performed with SYBR-Green PCR Mastermix (CAT:11202ES08). The expression levels of the tested genes were estimated using the relative $2^{-\triangle\triangle CT}$ method [24].

## RNA-Seq analysis and bioinformatics

The samples for RNA-Seq included the 2-week-old seedlings treated with 1 μM GA$_3$ and 20 μM ABA [25, 26]. The seedling were grown under hydroponics, the 1 μM GA3 and 20 μM ABA treatments through adding the appropriate amount of powder to the culture solution to achieve the required concentration. Three varieties of seedlings from the same provenance were mixed together and frozen in liquid nitrogen. There were three biological replicates per treatment and three samples per replicate. The RNA extractions and RNA-Seq analysis of the leaves from *F. hodginsii* followed previous methods [27].

RNA-Seq was performed using the Illumina NovaSeq 6000 with 6G of data and Poly (A) RNA from 1 mg total RNA or purified mRNA and purified m6A-containing fragments were

used to generate the cDNA libraries, respectively, according to TruSeq RNA Sample Prep Kit protocol. The sample library type was a eukaryotic unstranded specific transcriptome library. The trimming of raw data resulted in only the uniquely aligned reads, and a fold-change in expression of >1.3 and a false-discovery rate (FDR) <0.01 were set as the thresholds for designating the differentially expressed genes [28].

For the raw reads obtained from transcriptome sequencing, cut adapt and Perl scripts were used to remove splice sequences as well as low-quality sequences with a length <100 bp, <5% unknown bases, and bases with a quality value Q≤30 accounting for more than 20% of the whole sequence in the raw data, followed by verification of data quality using FASTQ. Analysis without a reference transcriptome was used, as no reference genome was available or the genome annotation information was incomplete.

The unigenes were annotated and classified according to GO (Gene Ontology) [2]. Based on this, differentially expressed genes were identified and subjected to GO term enrichment analysis. All samples were filtered using the transcriptome assembly software Trinity (Broad Institute, USA), and the resulting clean reads were assembled from scratch to obtain long fragments (contigs), which were assembled into sequences that could not be further extended at both ends, which were then finally normalized to unigenes. The unigenes were then evaluated for assembly quality, including length, GC content (GC%), and N50 [29].

The obtained unigenes were annotated against GO for functional annotation and classification, and differential expression analysis was performed on unigenes from different groups to find differentially expressed genes among the samples from the different localities of *F. hodginsii*.

## Statistical analyses

All data are from biological replicates. Data were analysed using one-way analysis of variance (ANOVA) tests (Tukey's post-hoc test), * and ** indicate significant differences (P<0.05 and P<0.01, respectively). Pairwise comparison was performed by Welch's t-test and different upper and lowercase letters indicated the variability analysis of the amount of change in the same object at different times. GraphPad v.8 was used to perform statistical analysis of the data.

## Results

### Phenotype of *F. hodginsii* from four provenances

The phenotypic data of *F. hodginsii* from four provenances collected at roughly the same latitude were analyzed (Fig 1A), and thus the annual rainfall and average temperature were also approximately the same. Based on phenotypic observation and data analysis, we found that *F. hodginsii* from Gutian was significantly different from the other provenances. The average clear height of *F. hodginsii* from P1, P2, P3, and P4 was 4.31 m, 4.36 m, 2.58 m, and 9.45 m, respectively, with P4 being the tallest and also significantly different from the other three. In terms of diameter at breast height (DBH), P1 was 26.65 cm, P2 was 29.16 cm, P3 was 30.18 cm, and P4 was 52.56 cm. P4 also had the greatest DBH and was highly significantly different from the others. In terms of branching angle, P4 was 72.53˚ and was smaller than the other three, which were 85.65˚, 82.14˚, and 73.75˚. There was a highly significant difference between P1 and P4 and a significant difference between P2 and P4, but no significant difference was detected between P3 and P4. In general, these features are consistent with flood tolerance (Fig 1B).

### Changes in endogenous ABA and GA$_3$ of *F. hodginsii* under flooding stress

To verify whether the endogenous ABA and GA$_3$ in *F. hodginsii* changed in response to flooding stress, the changes in ABA and GA$_3$ in wild-type *F. hodginsii* (Fuzhou local provenance)

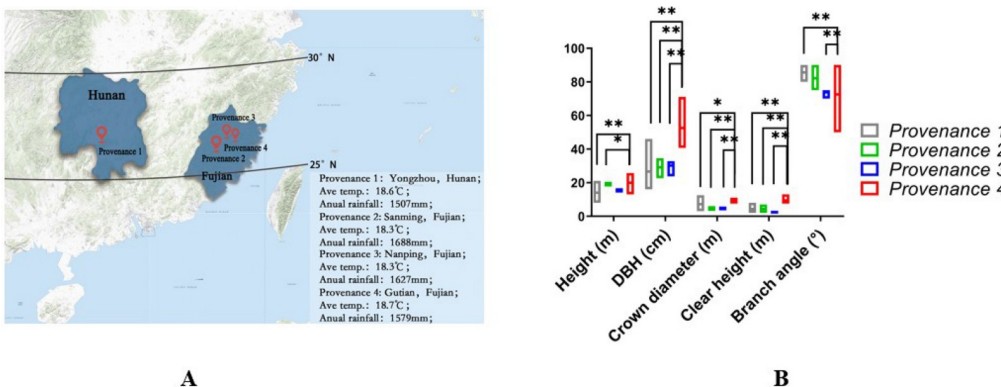

**Fig 1. Geographical distribution and phenotypic differences among *F. hodginsii* from four provenances.** (**A**) The geographical distribution of the four provenances, almost at the same latitude. Provenance 1 is from Yongzhou, Hunan (average temperature 18.6°C, annual rainfall 1507 mm); provenance 2 is from Sanming, Fujian (average temperature 18.3°C, annual rainfall 1688 mm); provenance 3 is from Nanping, Fujian (average temperature 18.3°C, annual rainfall 1627 mm); provenance 4 is from Gutian, Fujian (average temperature 18.3°C, annual rainfall 1579 mm). This picture is from USGS National Map Viewer (https://apps.nationalmap.gov/viewer/). (**B**) The phenotypic characteristics of the four provenances were statistically analyzed based on height (m), diameter at breast height (cm), crown diameter (m), clear height (m), and branch angle (°). Height is measured as the distance from the root on the ground to the top of the tree. DBH measures the diameter of a tree 1.3-m above the ground. Crown diameter is the arithmetic mean of the maximum and minimum numbers. Clear height calculates the height of the trunk. Branch angle represents the angle of the outermost branch of the crown. Data for each index were collected from 15 samples. * and ** indicate significant differences ($P<0.05$ and $P<0.01$, respectively) relative to P4 in height, diameter at breast height, crown diameter, clear height, and branch angle, based on one-way ANOVA with multiple comparisons using Tukey-test.

were detected by water flooding at four time points of 6 h, 12 h, 24 h, and 72 h. We found that the ABA content gradually decreased and the $GA_3$ content gradually increased with time in the wild type after 6 h of flooding treatment (Fig 2).

On the basis of these observations, we quantified the endogenous ABA and $GA_3$ contents of the three varieties of *F. hodginsii* from Gutian (G007, G008, G011) and the three varieties from Yongzhou (E010, E015, E025), as these two provenances exhibited significant phenotypic differences. The quantitative analysis revealed that the endogenous ABA content of all three varieties of *F. hodginsii* from Gutian was lower than that of the three varieties from Yongzhou, while in terms of $GA_3$ content, all three varieties from Gutian had higher contents than the three varieties of Yongzhou ($P<0.05$) (Fig 3A and 3B).

Next, the six varieties were subjected to flooding treatments for 6 h, 12 h, 24 h, and 72 h to detect the changes in endogenous ABA and $GA_3$. The results showed that the endogenous ABA content of both provenances showed a decreasing trend with the duration of the flooding treatment, but the three varieties from Gutian changed more rapidly, and the content of endogenous ABA was obviously less than in the three varieties from Yongzhou after 6 h. In terms of the changes in endogenous $GA_3$, the three varieties from Yongzhou exhibited no significant change and only increased slightly after 24 h of flooding treatment. However, the three varieties from Gutian changed rapidly, reaching a plateau stage after 12 h of flooding treatment, with a relative GA amount that was 3.25 times higher than that in Yongzhou (Fig 3C and 3D).

## Transcriptomic analyses reveal different ABA- and GA-related response genes

The transcriptomes of three varieties of *F. hodginsii* from P4-Gutian (G007, G008, G011) were compared with three varieties from P1-Yongzhou (E010, E015, E025), and a total of 6281

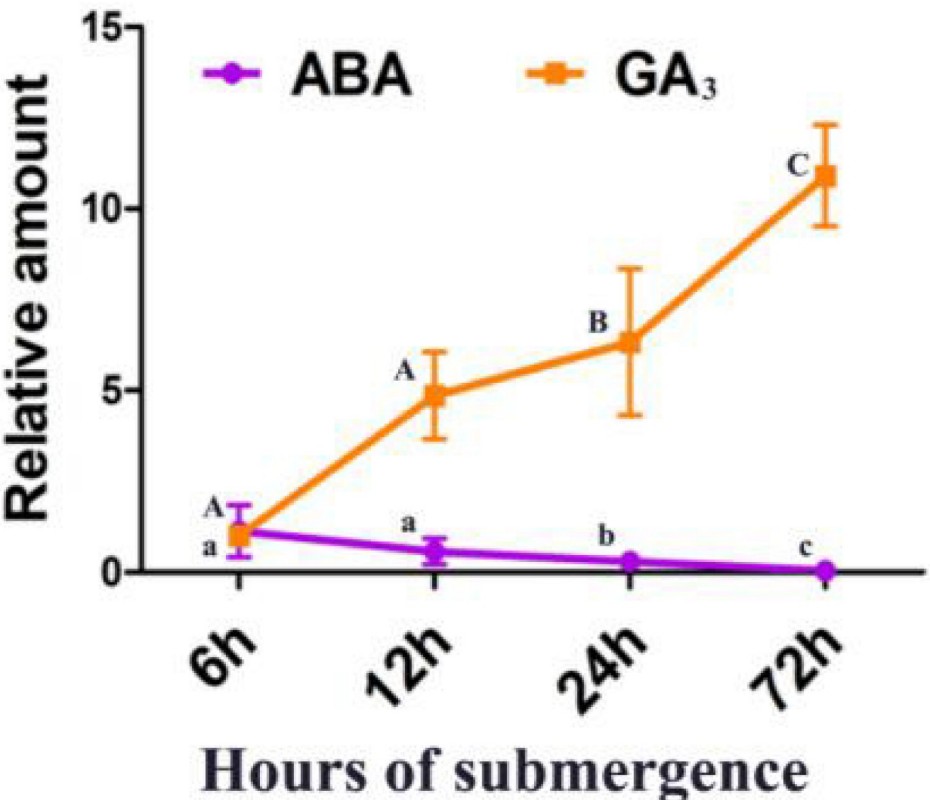

**Fig 2. The flooding stress treatments increased endogenous GA$_3$ and decreased endogenous ABA in the *F. hodginsii* seedings.** The tested seedings were local varieties of Fuzhou and were cultivated hydroponically for 2 weeks. The hydroponic solution was 10 g nutrient solution mixed with 1 L water. For the flooding stress treatments, the seedings were submerged in water for 6 h/12 h/24 h/72 h, the leaves were collected, and the endogenous ABA and GA$_3$ were determined. Three independent experiments were performed, each with three biological replicates. Data are the means of three biological repetitions ± SD, n = 9. Different capital letters (A, B, C) indicate the significance of GA$_3$ content at different treatment times compare to 6 h (p < 0.05), while different lowercase letters (a, b, c) indicate the significance of ABA content at different treatment times relative to 6 h (p < 0.05), Tukey-test was used for comparison between data.

differentially expressed genes were screened. There were 366 upregulated genes and 227 downregulated genes, accounting for 5.8% and 3.6%, respectively. GO functional analysis was performed by identifying the 15 functional categories with the most significant enrichment according to the upregulated and downregulated genes.

In the GO enrichment analysis of differentially upregulated genes in P4 compared to P1, 10 signaling pathways were identified as ABA-related genes (*P*<0.001) and were mainly related to biosynthesis, mitosis, and hormonal response. Among the upregulated genes, genes in the regulation of mitotic spindle organization and activation of protein kinase activity had a gene ratio of 0.75, which was the highest (Fig 4A). Eleven signaling pathways which were GA-related genes were detected in the GO enrichment analysis of the upregulated genes and were mainly involved in the metabolic and catabolic activities of life and the redox of cells and hormonal response, and the glycine catabolic process had the highest gene ratio of 0.6 (Fig 4B).

Eight related genes were identified by screening overlapping upregulated and downregulated genes in the regulation of salicylic acid mediated, cellular response to salicylic acid stimulus, and regulation of defense response signaling pathways, as indicated in S2 Table. Through designing primers for these eight genes and RT-qPCR, two genes with significant changes in

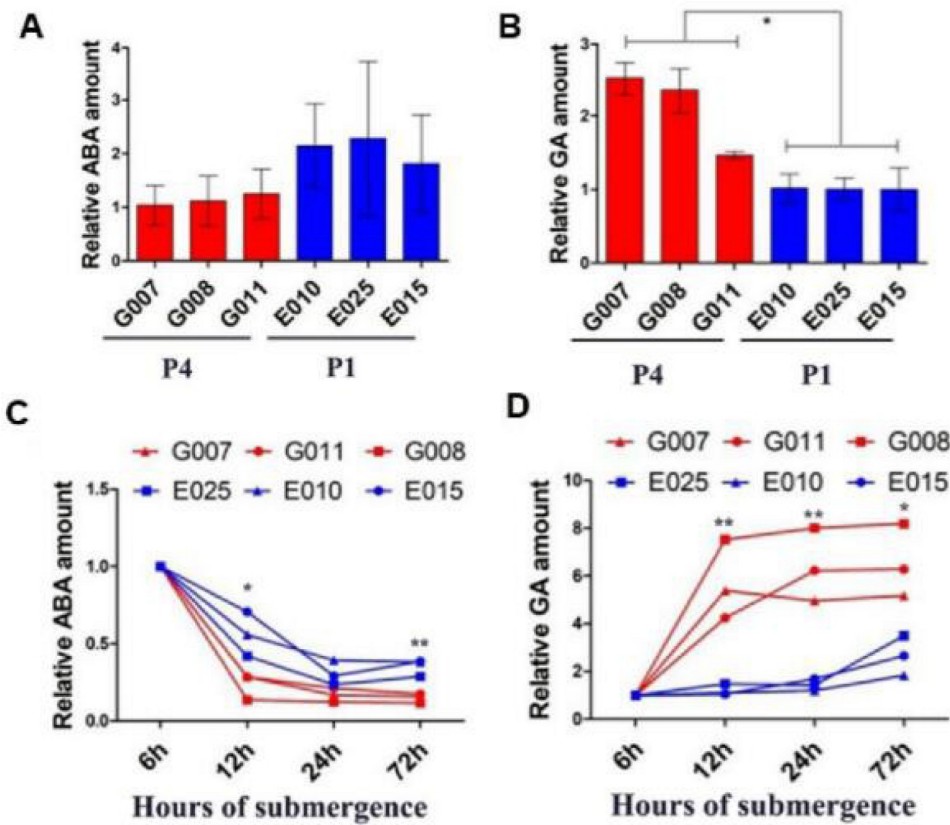

**Fig 3. Quantification of endogenous GA₃ and ABA in *F. hodginsii* and changes in response to flooding stress treatments in provenance 1 (P1) and provenance 4 (P4).** (**A&C**) Three varieties of P1 (E010, E025, E015) and P4 (G007, G008, G011) showing the amount of endogenous ABA indicating the change trend after 12 h/24 h/72 h flooding stress treatments compared with 6 h. (**B&D**) The amount of endogenous GA₃ in E010, E025, E015, and G007, G008, G011, and the change in GA₃ in the seedings subjected to 12 h/24 h/72 h flooding stress treatments compared with 6 h. The data are shown as means ± SD (n = 3). * and ** indicate significant differences (*P*<0.05 and *P*<0.01, respectively) between P1 and P2 in different times, based on one-way ANOVA with multiple comparisons using Tukey-test.

the regulation of ABA and GA₃ responses were selected from the quantification results. After 20 μM ABA treatment for 24 h, the expression of TRINITY_DN142_c0_g2 and TRINI-TY_DN7657_c0_g1 in P4 was lower than in P1 and there was a highly significant difference (Fig 4C). Following treatment with 1 μM GA₃ for 24 h, theTRINITY_DN142_c0_g2 and TRI-NITY_DN7657_c0_g1 genes of P1 were more highly expressed than P4 and were highly significantly different (Fig 4D).

## Discussion

Land plants are sensitive to waterlogging, which hinders the growth and survival [30, 31]. Numerous studies have reported that physiological and molecular alterations occur in plants under flooding conditions [32–34]. RNA-Seq is an effective method for assessing transcriptomic changes under waterlogging and can assist in elucidating waterlogging tolerance mechanisms in plants.

Previous studies found that *F. hodginsii* from Gutian possessed some phenotypes related to resistance to flooding, and the phenotypic characteristics of resistance to flooding were also more pronounced than in the other three provenances (Yongzhou, Sanming, Nanping).

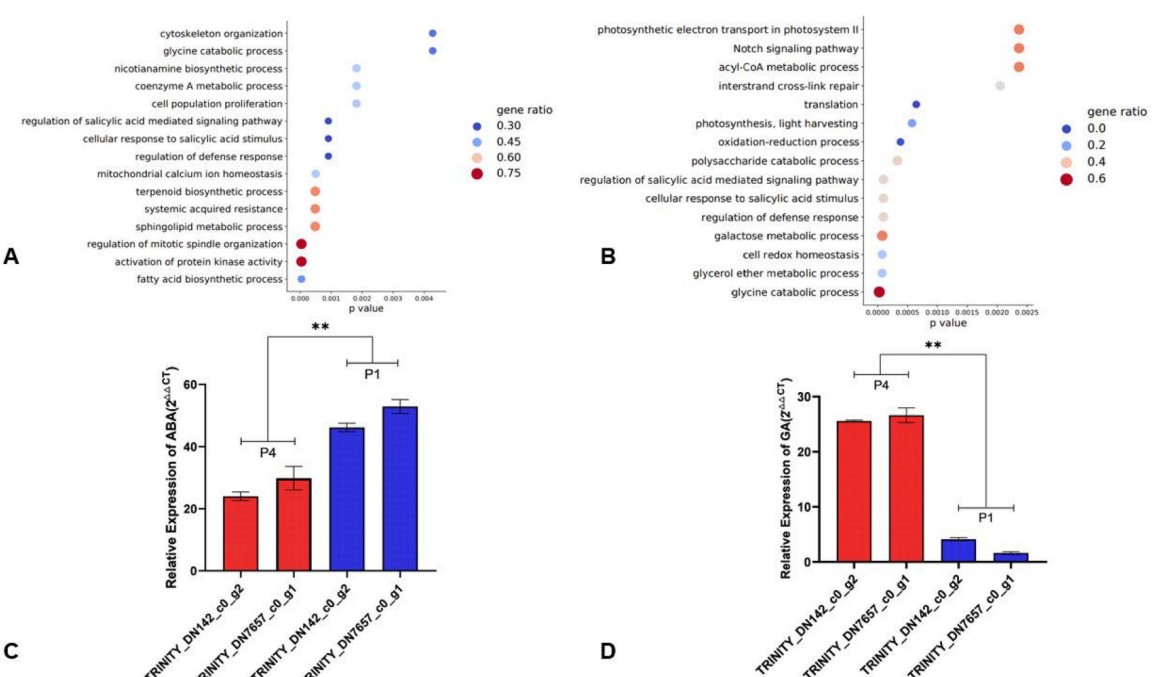

**Fig 4. GO term enrichment analysis and screening of GA₃ and ABA related major regulatory genes.** (**A**) and (**B**) 6281 differentially expressed genes were screened ($P<0.05$, fold-change $\geq$ 1) via the Illumina RNA-Seq results of P1 (E010, E025, E015 from provenance 1) and P4 (G007, G008, G011 from provenance 4). The RNA-Seq samples (n = 9) originated from hydroponically cultivated 2-week-old seedings, which were treated with 1 μM GA₃ and 20 μM ABA for 24 h. The GO analysis was based on the 15 regulatory pathways that had greater significant enrichment of upregulated and downregulated genes (arranged by *P*-value). The abscissa represents the *P*-value of the different genes in this functional classification. The ordinate represents different GO function classifications. The size of the dots represents this pathway gene number in relation to the total, and the color represents the enrichment. (**C**) and (**D**) Two crosstalk genes (TRINITY_DN142_c0_g2, TRINITY_DN7657_c0_g1) between ABA and GA₃ were selected from the regulation of salicylic acid-mediated, cellular response to salicylic acid stimulus, and regulation of defense response signaling pathway. P1 (E010, E025, E015) and P4 (G007, G008, G011) were treated with flooding stress for 24 h, and the samples from each provenance were combined. The relative expression of the crosstalk genes was indicated via qPCR ($2^{-\triangle\triangle CT}$). The data are shown as means ± SD (n = 3). ** indicates significant differences ($P<0.01$) based on one-way ANOVA with multiple comparisons using Tukey-test.

Genetic diversity analysis of P1, P2, P3, and P4 by SSRs revealed that P4 also had higher genetic diversity than the other three provenances. This suggested that P4 was more adaptable than P1, P2, and P3.

Phytohormones play a central role in flooding stress conditions for plant survival including morphological, anatomical, biochemical, molecular, and signaling mechanisms. GAs and ABA have crosstalk roles during submergence stress in plants [35], whereby ABA biosynthesis is reduced and GA signaling is induced [13]. The endogenous hormone contents and the associated change trends were explored by conducting flooding treatments in three varieties of P1 (E010, E015, E025) and P4 (G007, G008, G011). We found that the ABA content in G007, G008, and G011 was lower than in E010, E015, and E025. In terms of the content of GA₃, G007, G008, and G011 were all higher than E010, E015, and E025. Additionally, G007, G008, and G011 exhibited more visible trends than E010, E015, and E025 in terms of both ABA and GA₃. This is consistent with previous findings [14]. Based on these results, we infer that the degree of flooding tolerance is more prominent and the response to flooding stress is more dramatic in *F. hodginsii* from Gutian.

Here, 15 regulatory pathways that had greater significant enrichment of upregulated and downregulated genes which were arranged by P-value were identified using RNA-Seq and mainly related to biosynthesis, mitosis, the metabolic and catabolic activities of life, the redox

of cells and hormonal response. Three of these signaling pathways are present in both up- and down-regulated gene expression, they were regulation of salicylic acid-mediated, cellular response to salicylic acid stimulus and regulation of defense response, which were closely related to ABA and $GA_3$.

Many adaptation mechanisms are related to changes in ABA and GA metabolism and signaling [36–38]. ABA has a main role in regulating the stomata by adjusting the size of the guard cells and regulating the water potential in plants. Thus, ABA is considered to be a key hormone in water stress responses [39, 40]. ABA can be perceived in the guard cells, and ABA signaling can lead to a reduction in the turgor and volume of the guard cells via the efflux of anions and potassium ions and the gluconeogenic conversion of malate into starch, which finally results in stomatal closure; a process that has been previously well illustrated [41, 42]. GAs is one of the essential plant hormones regulating growth and development. GAs regulates multiple processes in plant growth and development, mainly by controlling the size and number of cells [43]. These functional modulations allow them to play an important role in plant defense response.

Ethylene (ET) is a gaseous hormone and its accumulation is an important way to response to flooding stress. It has long been acknowledged as the main regulator of plants' responses to reduced $O_2$ conditions [44–46]. Numerous studies show that ET signaling uncouples the catabolism of ABA, and constitutes a putative independent hypoxia signaling pathway [47]. Moreover, any effect on ABA signaling affects GA signaling, as ABA promotes the stabilization of the negatively regulated proteins of DELLA [48]. However, none of the above signaling pathways are associated with ethylene. Possible reasons are that ethylene is a gaseous hormone, which is volatile in air, and that it accumulates more in plant roots and fruits and less in leaves [49].

In addition to the plant defense response, two other signaling pathways were associated with salicylic acid (SA). In plants, SA is a common phenolic compound, which regulates cells' antioxidant mechanism through inducing the expression of stress-related genes to resist to stress [50–52]. Some experiments have shown that Salicylic acid, as a signal substance, can induce changes in physiological characteristics by increasing the activities of ethanol dehydrogenase, proline, POD and CAT. Thus, protecting leaves and root membranes from damage and maintaining photosynthesis in leaves as well as the activities of root [53]. Previous studies have investigated the function of hormones or salicylic acid as a single component in response to flooding stress, these signaling pathways provide a reference and reference for multi-omics studies of flooding stress.

Based on the above signaling pathway, we obtained eight related genes by screening for overlapping upregulated and downregulated genes from the three signaling pathways. Two genes, including TRINITY_DN142_c0_g2 and TRINITY_DN7657_c0_g1, which exhibited significant changes in the regulation of ABA and GA responses, were selected based on the results of the RT-qPCR. Following treatment with 1 μM $GA_3$ and 20 μM ABA for 24 h, the relative expression of these two genes in *F. hodginsii* of P1 was significantly higher than in P4. Treatment with 20 μM ABA resulted in the opposite. This indicated that *F. hodginsii* from Gutian had a better ability to adapt to the flooding stress. The TRINITY_DN142_c0_g2 and TRINITY_DN7657_c0_g1 genes may be central to its flooding resistance and might also constitute the basis of its adaptive evolution.

## Conclusion

When plants are exposed to flooding stress, they initiate various defensive measures, such as stomatal closure, shoot growth, and petiole or internode elongation control under both escape

and quiescence strategies. In previous experiments, we found that the varieties of *F. hodginsii* that originated from Gutian were taller and had a greater DBH but smaller branching angle compared with three other provenances originating from Yongzhou, Sanming, and Nanping. These features were consistent with flood tolerance, suggesting that the *F. hodginsii* varieties originating from Gutian possessed typical flooding tolerance traits compared to the other three provenances. We then used HPLC technology to measure the dynamics in endogenous $GA_3$ and ABA content, which indicated that ABA decreased but $GA_3$ increased with time under flooding stress. Furthermore, the $GA_3$ and ABA contents of the varieties originating from Gutian and the other three provenances were measured, which indicated that the *F. hodginsii* from Gutian could respond more rapidly to flooding stress. We then immediately compared the transcriptomes of the varieties originating from Gutian with the other three provenances via RNA-Seq. We identified eight related genes by screening for overlapping upregulated and downregulated genes in the regulation of salicylic acid-mediated, cellular response to salicylic acid stimulus, and regulation of defense response signaling pathways. To explore the underlying genetic mechanisms of the flood-resistant phenotypes, based on the primers designed for these eight genes and RT-qPCR, two genes (TRINITY_DN142_c0_g2 and TRINITY_DN7657_c0_g1) with significant changes in the regulation of ABA and GA responses were selected from the quantification results. Both genes were highly related to ABA and GA response and thus may be the reasons for phenotypic variation of *F. hodginsii*. The analysis of phenotypic differences in *F. hodginsii* revealed the intrinsic molecular regulatory mechanisms and provided some reference value for resistance breeding and adaptive evolution.

## Supporting information

**S1 Fig. Genetic diversity and phylogenetic evolution analysis of four provenance.** SSR (Simple Sequence Repeat) molecular markers were used to predict the genetic diversity among provenances 1–4. Using the DNA of provenances 1–4 as templates, SSR 1–11 primers were used to amplify sequences via PCR. Based on the amplified sequences, alleles frequency of provenance 1–4 were explored. The different colors represent the length of fragment separation, and the dendrogram indicates the genetic evolutionary distance.
(TIF)

**S1 Table. List of primers for SSR and RT-qPCR used in this research.**
(DOCX)

**S2 Table. Overlapping genes of ABA and GA.**
(DOCX)

## Acknowledgments

We thank LetPub (www.letpub.com) for its linguistic assistance during the preparation of this manuscript.

## Author Contributions

**Conceptualization:** Jun Su.

**Formal analysis:** Jian Li.

**Investigation:** Shunde Su.

**Methodology:** Jun Su.

**Project administration:** Xiangyang Kang, Renhua Zheng.

**Resources:** Shunde Su.

**Software:** Qing Zhao.

**Supervision:** Jian Li, Xiangyang Kang, Renhua Zheng.

**Visualization:** Tengfei Zhu.

**Writing – original draft:** Tengfei Zhu.

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
