## [Decision Letter · Decision Letter 0]

4 Oct 2021

PONE-D-21-26095Transcriptome analysis of gibberellins and abscisic acid during the flooding response in Fokienia hodginsiiPLOS ONE

Dear Dr. Zhu,

Thank you for submitting your manuscript to PLOS ONE. After careful consideration, we feel that it has merit but does not fully meet PLOS ONE’s publication criteria as it currently stands. Therefore, we invite you to submit a revised version of the manuscript that addresses the points raised during the review process.

Please address the comments of all reviewers. In particular, please provide the accession number for your sequences from RNA-seq (which should be deposited to a database prior to resubmission of a revision of this manuscript, but can be made publicly available after acceptance). In addition, please include full details of statistical analyses in the Materials and Methods, and be aware that fold change values for relative expression are not normally distributed, and thus should be shown with 95% confidence intervals and statistical tests that do not assume normal distribution. ddCT values can be used as an alternative to fold changes to avoid this issue.

We look forward to receiving your revised manuscript.

Kind regards,

Frances Sussmilch

Academic Editor

PLOS ONE

Journal Requirements:

2. Thank you for submitting the above manuscript to PLOS ONE. During our internal evaluation of the manuscript, we found significant text overlap between your submission and the following previously published works, some of which you are an author.

https://onlinelibrary.wiley.com/doi/10.1111/jipb.12313

The text that needs to be addressed involves the Discussion section.

Please revise the manuscript to rephrase the duplicated text, cite your sources, and provide details as to how the current manuscript advances on previous work. Please note that further consideration is dependent on the submission of a manuscript that addresses these concerns about the overlap in text with published work.

" The "Eagle Plan" Young Talent Project of Fujian Province."

6. Please include a copy of Tables 1 and 2 which you refer to in your text.

Reviewers' comments:

Reviewer's Responses to Questions

**Comments to the Author**

1. Is the manuscript technically sound, and do the data support the conclusions?

Reviewer #1: Yes

Reviewer #2: No

Reviewer #3: Partly

2. Has the statistical analysis been performed appropriately and rigorously? 

Reviewer #1: Yes

Reviewer #2: No

Reviewer #3: Yes

3. Have the authors made all data underlying the findings in their manuscript fully available?

Reviewer #1: Yes

Reviewer #2: No

Reviewer #3: No

4. Is the manuscript presented in an intelligible fashion and written in standard English?

Reviewer #1: Yes

Reviewer #2: Yes

Reviewer #3: Yes

5. Review Comments to the Author

Reviewer #1: Very nice paper on flooding tolerance! The data support is very clear. My main comments are about the grammar and wording structure, and to especially state the research gap this study fills.

Line 33: For “other three provenances” it may be helpful to specify the country of origin within introduction of the species.

Line 66: Missing a statement on the aim of this study at the end of the introduction section. What is novel about this research? Has physiological and molecular characterizations of F. hodginsii not been done before?

Line 78: Is the flooding assay done with some light or complete darkness?

Line 173: I feel the title “Different ABA- and GA-related response genes” is too simple and can be more elaborated, like “Transcriptomic analyses reveal different…”

Line 181: Missing hyphen “ABA related genes”

Line 228: Needs space between “1(“

Line 228: The jump between paragraphs from GA to SUB1 does not connect with the first sentence. It is more clear when the next sentence mentions GA. This may not be necessary, but perhaps there can be a better transition between the paragraphs.

Line 236: By the end of the SUB1 paragraph, it does not clarify how the SUB1 work relates to the data obtained. Perhaps a sentence can be added linking the connections between the studies, or how the SUB1 data can help support the current data.

Line 255: Before you mention the measurements of ABA and GA, you can highlight why you performed this work, the purpose, or how the research is novel. This is interpreted but not clearly stated.

Fig 1A: The black text in the map is hard to read and the font is small in the box.

Fig 1B: Since the squares are most strongly stacked and slightly bigger than the other symbols, perhaps showing a bar or box & whisker graph might make the data points more clear and easier to interpret.

Fig 2: Missing axis title for hours of submergence.

Fig 3A & C: It would help to draw a line under the provenance 1 & 4 samples in the graph.

Fig 3B & D: Missing axis title for hours of submergence.

Table S1: FJB91-R format needs to be centered.

More experimental support of course would be great, such as functional validation of the RNA-seq dataset, but for this level of publication I would not request any additional experiments.

Reviewer #2: The manuscript entitled 'Transcriptome analysis of gibberellins and abscisic acid during the flooding response in Fokienia hodginsii' describes the transcriptional and hormonal response of a Cupresaceaous tree to a 72h submergence period.

There are several critical aspects that need clarification:

1) In the M&M section says F. hodginsii seedlings were grown in hydroponics and subsequently (2 weeks after germination) completely immersed in water. The same hydroponic medium was used? Tap or clean water (potentially leading to the dilution of nutrients and a nutritional imbalance)? Seedlings were transplanted to soil prior to submergence? This needs to be further elaborated. O2 levels need to be measured in these conditions.

2) RNAseq data has to be deposited in a publicly available repository (e.g. NCBI SRA or similar) before manuscript is accepted for publication or upon acceptance, not later.

3) Authors recorded phenotypic data from F. hodginsii accessions collected from four different locations, but do not provide data that link these phenotypic traits with improved or decreased tolerance to anoxia. Moreover, phenotype responses to hypoxia were not measured or considered.

4) Typical responses to hypoxia are missing (ADH, PDC, SUSY upregulation) as well as induction of ERFVII homologs. This needs to be clearly present in the manuscript.

5) Statistical analysis is missing in Figures 1, 2 and 3

6) RNAseq results need to be provided as supplementary data, including annotation and statistical analyses.

7) Effects of flooding on GA and ABA signaling pathways needs to be supported also by transcriptional data. What GA was/were measured? What internal standards were used as surrogates?

Major aspects:

Ethylene has long been acknowledged as the main regulator of plants' responses to reduced O2 conditions, therefore, it cannot be excluded from the discussion. Interactions of ET with GA and ABA pathways in hypoxic conditions have been recently discussed (González-Guzmán et al. 2021).

A proper phenotypic evaluation of accessions in response to hypoxic conditions is required.

The proposal that SA signaling is involved is interesting but needs certain experimental backup. What happens with SA levels? What happens with typical SA-linked responses such as PR1 homolog overexpression? Could authors provide a tentative close relative of TRINITY_DN142_c0_g2 and TRINITY_DN7657_c0_g1?

L214-217 'suspect' is not scientific. Authors must provide plausible explanations well-grounded on empirical results.

L228-236 SUB1A is a non-canonical ERFVII which is not regulated by the N-end rule pathway and is present in rice. Does F. hodginsii has any potential ortholog? If not, these is out of context here. Other canonical ERFVII are better candidates to link ET and ABA or GA signaling (e.g. Arabidopsis RAP2.3, for instance).

For the sake of transparency and to allow replicability of results, transcriptional results must be presented in full, including annotation RPKM and statistics (FDR corrected) as supplementary.

To confirm that low O2 responses were activated under the experimental conditions described, expression of ET/O2 signaling ortholog elements must be presented along with typical anoxia survival responses (expression and/or enzyme activity of ADH, PDC and SUSY). Likewise, signaling pathways for GAs and ABA also need to be presented (expression of ABA receptors, PP2Cs, SnRKs and ABA-responsive genes such as ABI5, DREB2A, RD29, etc...).

Minor aspects:

L190 Where is Table 2, maybe authors refer to supporting Table S2. Please, amend.

Please, use the same coloring in Figures 3 and 4, as it stands is misleading.

Reviewer #3: Dear Dr. Tengfei Zhu,

I have read through the manuscript entitled "Transcriptome analysis of gibberellins and abscisic acid during the flooding response in Fokienia hodginsii”. The manuscript can provide new information in understanding the flooding tolerance mechanisms in non-model crops. With that in mind I would like to give the following comments and questions regarding the manusript.

1. Components of the PCR reaction volume doesn’t add up to 20. list them all. (Line 89)

Used GA standard/s (if used) not specified. There are various forms of GA available in the plants system, which one did you measure? (Line 95-102)

2. The gel electrophoresis will not tell you much/precisely the quality of the RNA. Have you run a the samples on the bioanalyzer? or Nano drop absorbance for these purposes? (Line 105)

3. what reference genome did you use to design the primers? (Line 109)

which qRT-PCR instrument did you use? (Line 108-113)

4. consistency in writing terminologies. For example qRT-PCR (Line 108) vs RT-qPCR (supporting information line 9. Look into the rest of the manuscript for consistency

5. How did you calculate the relative GA and ABA amounts in Fig. 2 and 3 (Line 14-19; 22-28)? it should be elaborated in the figure legends and method section.

6. I recommend to include the hormone analysis data, at least in the supporting information section of the manuscript.

7. Can the authors reanalyse the transcriptomic data considering the quality check used at the begining of the analysis were Q10. It should be at least Q30 with inferred base call accuracy of 99.99% rather than 90% in Q10 (Line 124).

8. How do you explain that you have supplemented the plants to be analyzed for RNAseq with GA3 but quantification from the other set of experiment stated only endogeneous GA. Can you reanalyze the hormone data by including other GA forms rather than stating indogenoeus GA, considering their que-specific responses?

Regards,

6. PLOS authors have the option to publish the peer review history of their article (what does this mean?). If published, this will include your full peer review and any attached files.

Reviewer #1: **Yes: **Elaine Yeung

Reviewer #2: No

Reviewer #3: No

---

## [Author Response · Author response to Decision Letter 0]

29 Nov 2021

Journal Requirement:

1. Please address the comments of all reviewers. In particular, please provide the accession number for your sequences from RNA-seq (which should be deposited to a database prior to resubmission of a revision of this manuscript, but can be made publicly available after acceptance).

Response: The sequence data reported in this study have been deposited in the Genome Sequence Archive in BIG Data Center, Beijing Institute of Genomics (BIG), Chinese Academy of Sciences, under accession numbers CRA005262, CRA005262 that are publicly accessible at https://bigd.big.ac.cn/gsa/browse/CRA005262.

2. Please include full details of statistical analyses in the Materials and Methods.

Response: Line 175. We have added the statistical analyses in the Materials and Methods.

3. The text that needs to be addressed involves the Discussion section.

Response: We have reworked the sentences in the discussion section.

" The "Eagle Plan" Young Talent Project of Fujian Province."

Response: We have amended statements in cover letter.

Response: No ethics statement need to claim in this manuscript.

6. We note that Figure 1 in your submission contain [map/satellite] images which may be copyrighted. All PLOS content is published under the Creative Commons Attribution License (CC BY 4.0), which means that the manuscript, images, and Supporting Information files will be freely available online, and any third party is permitted to access, download, copy, distribute, and use these materials in any way, even commercially, with proper attribution. For these reasons, we cannot publish previously copyrighted maps or satellite images created using proprietary data, such as Google software (Google Maps, Street View, and Earth).

Response: we have supplied a replacement figure that complies with the CC BY 4.0 license from USGS National Map Viewer (public domain): http://viewer.nationalmap.gov/viewer/.

Response to reviewer:

Reviewer #1.

1. Line 33: For “other three provenances” it may be helpful to specify the country of origin within introduction of the species.

Response: Line 27. This is a very good suggestion. We have already added the regions of origin to specify “other three provenances”.

2. Line 66: Missing a statement on the aim of this study at the end of the introduction section. What is novel about this research? Has physiological and molecular characterizations of F. hodginsii not been done before?

Response: Line 74. Thank you for reminding. We have added the statement at the end of introduction.

As far as I know, the research on F. hodginsii is mainly focused on genetic evolution and metabolites, and there is very little gene function research. 

3. Line 78: Is the flooding assay done with some light or complete darkness?

Response: Line91. It was conducted in complete darkness to minimize the o2- contamination and we have added the condition of this treatment. 

4. Line 173: I feel the title “Different ABA- and GA-related response genes” is too simple and can be more elaborated, like “Transcriptomic analyses reveal different…”

Response: Line 222. Thanks for this kind advice, we have changed the title “Different ABA- and GA-related response genes” to “Transcriptomic analyses reveal different ABA- and GA-related response genes”.

5. Line 181: Missing hyphen “ABA related genes”

Response: Line 230. We deeply sorry about this mistake, we have added the hyphen.

6. Line 228: Needs space between “1(“

Response: This less relevant paragraph has been removed based on the comments of other reviewers.

7. Line 228: The jump between paragraphs from GA to SUB1 does not connect with the first sentence. It is more clear when the next sentence mentions GA. This may not be necessary, but perhaps there can be a better transition between the paragraphs.

Response: This less relevant paragraph has been removed based on the comments of other reviewers.

8. Line 236: By the end of the SUB1 paragraph, it does not clarify how the SUB1 work relates to the data obtained. Perhaps a sentence can be added linking the connections between the studies, or how the SUB1 data can help support the current data.

Response: This less relevant paragraph has been removed based on the comments of other reviewers.

9. Line 255: Before you mention the measurements of ABA and GA, you can highlight why you performed this work, the purpose, or how the research is novel. This is interpreted but not clearly stated. 

Response: Line 49. In the background section, we have added an elaboration on the selection of ABA and GA.

Fig 1A: The black text in the map is hard to read and the font is small in the box.

Response: Corrections have been made.

Fig 1B: Since the squares are most strongly stacked and slightly bigger than the other symbols, perhaps showing a bar or box & whisker graph might make the data points more clear and easier to interpret.

Response: Corrections have been made.

Fig 2: Missing axis title for hours of submergence.

Response: Corrections have been made.

Fig 3A & C: It would help to draw a line under the provenance 1 & 4 samples in the graph.

Response: Corrections have been made.

Fig 3B & D: Missing axis title for hours of submergence.

Response: Corrections have been made.

Table S1: FJB91-R format needs to be centered.

Response: Corrections have been made.

Reviewer #2.

1. In the M&M section says F. hodginsii seedlings were grown in hydroponics and subsequently (2 weeks after germination) completely immersed in water. The same hydroponic medium was used? Tap or clean water (potentially leading to the dilution of nutrients and a nutritional imbalance)? Seedlings were transplanted to soil prior to submergence? This needs to be further elaborated. O2 levels need to be measured in these conditions.

Response: Line 89. F. hodginsii seedlings were cultured used the same hydroponic medium and the water used treated was purified water, these have been added in the M&M section. About O2, Flooding is a complex abiotic stress, anoxia is one of the important stress caused by flooding, it is reasonable to clarify the O2 levels in different conditions. But in our case, we want provide a basic understanding of the flooding, the integrated stress, other than part of it. Moreover, O2 level was not always mentioned in similar works, partly because quantify the O2 levels may miss lead the audience that anoxia can be isolated from the flooding stress. 

2. RNAseq data has to be deposited in a publicly available repository (e.g. NCBI SRA or similar) before manuscript is accepted for publication or upon acceptance, not later.

Response: The sequence data reported in this study have been deposited in the Genome Sequence Archive in BIG Data Center, Beijing Institute of Genomics (BIG), Chinese Academy of Sciences, under accession numbers CRA005262, CRA005262 that are publicly accessible at https://bigd.big.ac.cn/gsa/browse/CRA005262.

3. Authors recorded phenotypic data from F. hodginsii accessions collected from four different locations, but do not provide data that link these phenotypic traits with improved or decreased tolerance to anoxia. Moreover, phenotype responses to hypoxia were not measured or considered.

Response: Thanks for your advices. As we mentioned earlier, anoxia is one of the stress caused by flooding, not all of it. And, it was well studied that controlling of shoot, petiole, or internode elongation were the phenotypes greatly associated with flooding stress. Moreover, it is impossible to collect phenomics data other than height, diameter at breast height, crown diameter, clear height, and branch angle in the field, considering the complex working condition and poor operability. In this work, phenotypic data from the field is only the hint to leading this research. The GA and ABA content in different conditions were the real “phenotypic data” to explain the transcriptome results. 

4. Typical responses to hypoxia are missing (ADH, PDC, SUSY upregulation) as well as induction of ERFVII homologs. This needs to be clearly present in the manuscript.

Response: The RNA-seq of F. hodginsii was no reference transcriptome, the information of the proteins available is limited, so we do not have the conditions to explore these target proteins, and we can analyze them according to our current results when the reference genome information is released.

5. Statistical analysis is missing in Figures 1, 2 and 3

Response: Thanks, corrections have been made.

6. RNAseq results need to be provided as supplementary data, including annotation and statistical analyses.

Response: Sequencing analysis and annotations have been uploaded to the appendix.

7. Major aspects:

Ethylene has long been acknowledged as the main regulator of plants' responses to reduced O2 conditions, therefore, it cannot be excluded from the discussion. Interactions of ET with GA and ABA pathways in hypoxic conditions have been recently discussed (González-Guzmán et al. 2021).

Response: Line 283. We have added the interactions of ET with GA and ABA pathways in hypoxic conditions in the discussion section.

A proper phenotypic evaluation of accessions in response to hypoxic conditions is required.

Response: Flooding is a complex abiotic stress, anoxia is one of the important stress caused by flooding, it is reasonable to clarify the O2 levels in different conditions. But in our case, we want provide a basic understanding of the flooding, the integrated stress, other than part of it. Moreover, O2 level was not always mentioned in similar works, partly because quantify the O2 levels may miss lead the audience that anoxia can be isolated from the flooding stress.

The proposal that SA signaling is involved is interesting but needs certain experimental backup. What happens with SA levels? What happens with typical SA-linked responses such as PR1 homolog overexpression? Could authors provide a tentative close relative of TRINITY_DN142_c0_g2 and TRINITY_DN7657_c0_g1?

Response: Line 291. Thank you for the affirmation. We have added in the Discussion section about the functions played by SA in abiotic stresses and future research will also be conducted to explain this issue.

L214-217 'suspect' is not scientific. Authors must provide plausible explanations well-grounded on empirical results.

Response: Line 265. Thank you for the advice, we have changed the 'suspect' to 'infer'.

L228-236 SUB1A is a non-canonical ERFVII which is not regulated by the N-end rule pathway and is present in rice. Does F. hodginsii has any potential ortholog? If not, these is out of context here. Other canonical ERFVII are better candidates to link ET and ABA or GA signaling (e.g. Arabidopsis RAP2.3, for instance). 

Response: Indeed, as you said, this paragraph was not convincing, so we have removed it.

For the sake of transparency and to allow replicability of results, transcriptional results must be presented in full, including annotation RPKM and statistics (FDR corrected) as supplementary.

Response: Thanks for reminding. Sequencing analysis and annotations have been uploaded to the appendix.

To confirm that low O2 responses were activated under the experimental conditions described, expression of ET/O2 signaling ortholog elements must be presented along with typical anoxia survival responses (expression and/or enzyme activity of ADH, PDC and SUSY). Likewise, signaling pathways for GAs and ABA also need to be presented (expression of ABA receptors, PP2Cs, SnRKs and ABA-responsive genes such as ABI5, DREB2A, RD29, etc...).

Response: Thanks for the suggestion. The transcriptome of this article is a non-reference transcriptome, compared to the reference transcriptome, functional gene analysis is not as detailed. Therefore, our idea is to screen as well as validate the F. hodginsii genes with the highest correlation with water flooding, and it is difficult to provide any other valuable information.

8. Minor aspects:

L190 Where is Table 2, maybe authors refer to supporting Table S2. Please, amend.

Please, use the same coloring in Figures 3 and 4, as it stands is misleading.

Response: We deeply sorry about the mistake. We have changed “Table 2” to “Table S2”.

Reviewer #3.

1. Components of the PCR reaction volume doesn’t add up to 20. list them all. (Line 89)

Used GA standard/s (if used) not specified. There are various forms of GA available in the plants system, which one did you measure? (Line 95-102)

Response: Line 101.Thanks for pointing this out. We have listed “7 µL ddH2O”.

 Line 107. We detected the contents of endogenous ABA and GA3, corrections have been made.

2. The gel electrophoresis will not tell you much/precisely the quality of the RNA. Have you run a the samples on the bioanalyzer? or Nano drop absorbance for these purposes? (Line 105)

Response: The gel electrophoresis is only to detect whether RNA is degraded, and we have measured the RNA concentration, absorbance value, 260/230, 260/280 before reverse transcription.

3. what reference genome did you use to design the primers? (Line 109)

which qRT-PCR instrument did you use? (Line 108-113)

Response: Line 145.primers were designed by using De novoassembly RNA-seq data (Table S1) and qRT-PCR instrument is Applied Biosystems QuantStudio™ 7 Flex. 

4. consistency in writing terminologies. For example qRT-PCR (Line 108) vs RT-qPCR (supporting information line 9. Look into the rest of the manuscript for consistency

Response: Thanks for pointing out, Line145,240,303,325 - we have standardized to “RT-qPCR".

5. How did you calculate the relative GA and ABA amounts in Fig. 2 and 3 (Line 14-19; 22-28)? it should be elaborated in the figure legends and method section.

I recommend to include the hormone analysis data, at least in the supporting information section of the manuscript.

Response: Line 109. In the Materials and Methods section, we have added detailed steps and methods for the determination.

6. Can the authors reanalyse the transcriptomic data considering the quality check used at the begining of the analysis were Q10. It should be at least Q30 with inferred base call accuracy of 99.99% rather than 90% in Q10 (Line 124).

Response: Line 162. Thanks for pointing out, we checked the sequencing results again, sequence splicing assembly is based on Q30 standard to ensure the accuracy of the results. 

7. How do you explain that you have supplemented the plants to be analyzed for RNAseq with GA3 but quantification from the other set of experiment stated only endogeneous GA. Can you reanalyze the hormone data by including other GA forms rather than stating indogenoeus GA, considering their que-specific responses?

Response: Thanks for pointing this out. Actually, we only quantified GA3 by HPLC-MS since GA3 was proved to be the most sensitive form of GA to flooding. Relative changes have been made to clarify that only GA3 was quantified in this study

---

## [Decision Letter · Decision Letter 1]

19 Dec 2021

PONE-D-21-26095R1Transcriptome analysis of gibberellins and abscisic acid during the flooding response in Fokienia hodginsiiPLOS ONE

Dear Dr. Zhu,

Thank you for submitting your manuscript to PLOS ONE. After careful consideration, we feel that it has merit but does not fully meet PLOS ONE’s publication criteria as it currently stands. Therefore, we invite you to submit a revised version of the manuscript that addresses the points raised during the review process.

As some former reviewers were unavailable to review the revised manuscript, a new reviewer (Reviewer 4) provided a review. Please address Reviewer 4’s comments. In addition, please address the following minor corrections (line numbers from track changes manuscript version):

L121 space before comma should be deletedL145 should be “DNase I”L184 CRA005262 is currently duplicated Please submit your revised manuscript by Feb 02 2022 11:59PM. If you will need more time than this to complete your revisions, please reply to this message or contact the journal office at plosone@plos.org. Please include the following items when submitting your revised manuscript:A rebuttal letter that responds to each point raised by the academic editor and reviewer(s). You should upload this letter as a separate file labeled 'Response to Reviewers'.A marked-up copy of your manuscript that highlights changes made to the original version. You should upload this as a separate file labeled 'Revised Manuscript with Track Changes'.An unmarked version of your revised paper without tracked changes. You should upload this as a separate file labeled 'Manuscript'.If applicable, we recommend that you deposit your laboratory protocols in protocols.io to enhance the reproducibility of your results. Protocols.io assigns your protocol its own identifier (DOI) so that it can be cited independently in the future. For instructions see: https://journals.plos.org/plosone/s/submission-guidelines#loc-laboratory-protocols. Additionally, PLOS ONE offers an option for publishing peer-reviewed Lab Protocol articles, which describe protocols hosted on protocols.io. Read more information on sharing protocols at https://plos.org/protocols?utm_medium=editorial-email&utm_source=authorletters&utm_campaign=protocols.

We look forward to receiving your revised manuscript.

Kind regards,

Frances Sussmilch

Academic Editor

PLOS ONE

Journal Requirements:

Reviewers' comments:

Reviewer's Responses to Questions

**Comments to the Author**

1. If the authors have adequately addressed your comments raised in a previous round of review and you feel that this manuscript is now acceptable for publication, you may indicate that here to bypass the “Comments to the Author” section, enter your conflict of interest statement in the “Confidential to Editor” section, and submit your "Accept" recommendation.

Reviewer #3: All comments have been addressed

Reviewer #4: (No Response)

2. Is the manuscript technically sound, and do the data support the conclusions?

Reviewer #3: Yes

Reviewer #4: Partly

3. Has the statistical analysis been performed appropriately and rigorously? 

Reviewer #3: Yes

Reviewer #4: Yes

4. Have the authors made all data underlying the findings in their manuscript fully available?

Reviewer #3: Yes

Reviewer #4: Yes

5. Is the manuscript presented in an intelligible fashion and written in standard English?

Reviewer #3: Yes

Reviewer #4: Yes

6. Review Comments to the Author

Reviewer #3: I have read the revised version of the manuscript entitled "Transcriptome analysis of gibberellins and abscisic acid during the flooding response in Fokienia hodginsii”. The manuscript is informative and can provide new insight in understanding the flooding tolerance mechanisms in non-model crops, mainly F. hodginsii.

Reviewer #4: 1. At no point in the text is cited the country where the provinces are located. It is implied but not explicit. Furthermore, the denominations of the provinces are ambiguous and change during the text, making the understanding of the reader from another country confusing.

2. In the introduction, in the paragraph beginning on line 71, the flooding conditions that may have triggered the plants' adaptation to excess water are mentioned. It would be interesting to inform the time needed for the studied species to complete a reproductive cycle, since the epigenetic changes triggered by stress might not have established themselves as marks transmitted between generations in a short period of time.

3. In lines 85-87 it is not clear whether the plants were grown under hydroponic conditions or in soil.

4. In line 91, it would be pertinent to add the justification for the growth to have occurred in the dark. Do the authors believe that the gene expression observed in the dark condition reflects the transcript pattern that would be expected for these plants under natural conditions? Furthermore, there is no information on acclimation to the dark condition or the growing temperature.

5. In line 100 there is no information on where the sequences of primers used in the experiment are found.

6. In line 122 there is not enough information about the analysis conditions in UHPLC.

7. In the “Simple Sequence Repeat (SSR) assay” section, there is no information on the methodology for extracting genetic material. How many grams of material were used? Did each sample come from different plants or from a pool of individuals? Couldn't the genetic variability of the species mask the results if different individuals make up the same sample?

8. In the section “Total RNA extraction, cDNA reverse transcription, and quantitative real-time (qRT) PCR” there is no information on the methodology used to extract the genetic material from the sample. How many grams were used in the extraction and which plant organs were collected for this trial? Did each individual compose a sample or was a pool of individuals performed during the extraction? Given the genetic variability of the species, would the pool represent a uniform condition of transcripts during the analyses?

9. In the section “RNA-Seq analysis and bioinformatics” an experiment is mentioned where the seedlings were treated with 1 µM GA3 and 20 µM ABA, but this methodology was not described in the text.

10. In line 159 the conditions for RNA sequencing were not described.

11. The “INITY_DN7657_c0_g1” gene is constantly written without “TR” (lines 36, 248, 250, 308 and 313 for example), and it was not clear if it was a typo or if the name really is “inity, since in the response to Reviewer #2 and in the caption of Figure 4 it was written as “TRINITY_DN7657_c0_g1”.

12. On line 291, correct O2 to O2

13. Between lines 289-296 the role of ethylene in flooding conditions was discussed. Did the authors analyze ethylene-responsive genes in transcriptomic analysis? The same can be pointed out in the discussion of salicylic acid (lines 297-305).

14. In the conclusion, it would be interesting to include the relevance of the results obtained in this work, as well as the future perspective for new experiments.

15. With regard to the images, in Figure 1B the data caption in the graph is in inverse order to the bars in the graphs (in the bars Province 4 appears first, although it is the last in the image caption).

16. It would be interesting to include the title of each graphic in the images (and not just in the caption).

17. Figures 4A and 4B are very small and difficult to read.

7. PLOS authors have the option to publish the peer review history of their article (what does this mean?). If published, this will include your full peer review and any attached files.

Reviewer #3: No

Reviewer #4: No

---

## [Author Response · Author response to Decision Letter 1]

19 Jan 2022

To editor:

1. L121 space before comma should be deleted

Respone: Line 123. Thank you for reminding. Corrections have been made.

2. L145 should be “DNase I”

Respone: Line 148. Thank you for reminding. Corrections have been made.

3. L184 CRA005262 is currently duplicated

Respone: Line 188. Thank you for reminding. Corrections have been made.

To reviewer:

1. At no point in the text is cited the country where the provinces are located. It is implied but not explicit. Furthermore, the denominations of the provinces are ambiguous and change during the text, making the understanding of the reader from another country confusing.

Response: Line79. Thank you for reminding. The original country “China” has been added for F. hodginsii. 

2. In the introduction, in the paragraph beginning on line 71, the flooding conditions that may have triggered the plants' adaptation to excess water are mentioned. It would be interesting to inform the time needed for the studied species to complete a reproductive cycle, since the epigenetic changes triggered by stress might not have established themselves as marks transmitted between generations in a short period of time.

Response: Line 67. “It takes about 5-7 years from germination to flowering, with flowering in March-April and fruiting in October-November” has been added. The time of the 1956 flooding was well documented, perhaps it happened before that too, but there is no record of it, so we have no way of knowing. And later seed gardens may have accelerated this genetic effect.

3. In lines 85-87 it is not clear whether the plants were grown under hydroponic conditions or in soil.

Response: Line87. “under hydroponics” has been added.

4. In line 91, it would be pertinent to add the justification for the growth to have occurred in the dark. Do the authors believe that the gene expression observed in the dark condition reflects the transcript pattern that would be expected for these plants under natural conditions? Furthermore, there is no information on acclimation to the dark condition or the growing temperature.

 Response: Line 92. “all treatments were performed under complete dark conditions that included 22℃ and 85% relative humidity.” has been added. It was conducted in complete darkness to minimize the O2- contamination. Dark conditions were operated in a darkened room which included 22℃ and 85% relative humidity.

5. In line 100 there is no information on where the sequences of primers used in the experiment are found.

Response: Line99. Thank you for reminding. “Eleven pairs of SSR primers were used to analyse [20] (Table S1)” has been added.

6. In line 122 there is not enough information about the analysis conditions in UHPLC.

 Response: Line 129. We have refined the analysis conditions of UHPLC-MRM-MS and “The MRM parameters for each of the targeted analytes were optimized using flow injection analysis, by injecting the standard solutions of the individual analytes, into the ESI source of the mass spectrometer. Several most sensitive transitions were used in the MRM scan mode to optimize the collision energy for each Q1/Q3 pair. Among the optimized MRM transitions per analyte, the Q1/Q3 pairs that showed the highest sensitivity and selectivity were selected as ‘quantifier’ for quantitative monitoring. The additional transitions acted as ‘qualifier’ for the purpose of verifying the identity of the target analytes. Skyline Software were employed for MRM data acquisition and processing.” has been added.

7. In the “Simple Sequence Repeat (SSR) assay” section, there is no information on the methodology for extracting genetic material. How many grams of material were used? Did each sample come from different plants or from a pool of individuals? Couldn't the genetic variability of the species mask the results if different individuals make up the same sample?

Response: Line 96. Approximately 100mg leaf samples were used to extract DNA. They were pool of individuals from same.

8. In the section “Total RNA extraction, cDNA reverse transcription, and quantitative real-time (qRT) PCR” there is no information on the methodology used to extract the genetic material from the sample. How many grams were used in the extraction and which plant organs were collected for this trial? Did each individual compose a sample or was a pool of individuals performed during the extraction? Given the genetic variability of the species, would the pool represent a uniform condition of transcripts during the analyses?

Response: Line151. We have added the detail “500 mg leaves”. The samples were pool of individuals from same.

9. In the section “RNA-Seq analysis and bioinformatics” an experiment is mentioned where the seedlings were treated with 1 µM GA3 and 20 µM ABA, but this methodology was not described in the text.

Response: Line 166. “The seedling were grown under hydroponics, the 1 µM GA3 and 20 µM ABA treatments through adding the appropriate amount of powder to the culture solution to achieve the required concentration.” has been added.

10. In line 159 the conditions for RNA sequencing were not described.

Response: Line 172. We have added the conditions for RNA-seq and “RNA-Seq was performed using the Illumina NovaSeq 6000 with 6G of data and Poly (A) RNA from 1 mg total RNA or purified mRNA and purified m6A-containing fragments were used to generate the cDNA libraries, respectively, according to TruSeq RNA Sample Prep Kit protocol. The sample library type was a eukaryotic unstranded specific transcriptome library.” has been added.

11. The “INITY_DN7657_c0_g1” gene is constantly written without “TR” (lines 36, 248, 250, 308 and 313 for example), and it was not clear if it was a typo or if the name really is “inity, since in the response to Reviewer #2 and in the caption of Figure 4 it was written as “TRINITY_DN7657_c0_g1”.

Response: Lines 36, 265, 267, 335, 330 and 349.Thank you for reminding. Corrections have been made.

12. On line 291, correct O2 to O2

Response: Line 308. Thank you for reminding. The “O2” has corrected to “O2”.

13. Between lines 289-296 the role of ethylene in flooding conditions was discussed. Did the authors analyze ethylene-responsive genes in transcriptomic analysis? The same can be pointed out in the discussion of salicylic acid (lines 297-305).

Response: These two relevant changes were not found in the available data. Due to genomic information is not available, there is no way to answer the above question further.

14. In the conclusion, it would be interesting to include the relevance of the results obtained in this work, as well as the future perspective for new experiments.

Response: Line 351. We describe the relevance of the results and the outlook and “thus may be the reasons for phenotypic variation of F. hodginsii. The analysis of phenotypic differences in F. hodginsii revealed the intrinsic molecular regulatory mechanisms and provided some reference value for resistance breeding and adaptive evolution.” has been added.

15. With regard to the images, in Figure 1B the data caption in the graph is in inverse order to the bars in the graphs (in the bars Province 4 appears first, although it is the last in the image caption).

Response: Thank you for reminding. We have corrected the figure.

16. It would be interesting to include the title of each graphic in the images (and not just in the caption).

Response: Thank you for your advice. We tried to take your suggestion, but some of the images have more characters and look a bit crowded with the captions.

17. Figures 4A and 4B are very small and difficult to read.

 Response: Sorry about this. We have changed the size and sharpness of the images.

---

## [Editor Report · Decision Letter 2]

21 Jan 2022

Transcriptome analysis of gibberellins and abscisic acid during the flooding response in Fokienia hodginsii

PONE-D-21-26095R2

Dear Dr. Zhu,

We’re pleased to inform you that your manuscript has been judged scientifically suitable for publication and will be formally accepted for publication once it meets all outstanding technical requirements.

Kind regards,

Frances Sussmilch

Academic Editor

PLOS ONE
---

## [Editor Report · Acceptance letter]

2 Feb 2022

PONE-D-21-26095R2 

Transcriptome analysis of gibberellins and abscisic acid during the flooding response in *Fokienia hodginsii*

Dear Dr. Zhu:

I'm pleased to inform you that your manuscript has been deemed suitable for publication in PLOS ONE. Congratulations! Your manuscript is now with our production department. 

Kind regards, 

on behalf of

Dr. Frances Sussmilch 

Academic Editor

PLOS ONE